# Phenotypic quantification of root spatial distribution along circumferential direction for field paddy-wheat

Xinxin Chen[1], Yongli Tang[2], Qingfei Duan[3], Jianping Hu[1]*

**1** School of Agricultural Engineering, Jiangsu University, Zhenjiang, China, **2** Nanjing Agricultural Equipment Extension Center, Nanjing, China, **3** College of Engineering, Nanjing Agricultural University, Nanjing, China

* hujp@ujs.edu.cn

## Abstract

Plant roots are essential for water and nutrient absorption, anchoring, mechanical support, metabolite storage and interaction with the surrounding soil environment. A comprehensive understanding of root traits provides an opportunity to build ideal roots architectural system that provides improved stability and yield advantage in adverse target environments caused by soil quality degradation, climate change, etc. However, we hypothesize that quantitative indicators characterizing root system are still need to be supplemented. Features describing root growth and distribution, until now, belong mostly to 2D indicators or reflect changes in the root system with a depth of soil layers but are rarely considered in a spatial region along the circumferential direction. We proposed five new indicators to quantify the dynamics of the root system architecture (RSA) along its eight-part circumferential orientations with visualization technology which consists of in-situ field root samplings, RSA digitization, and reconstruction according to previous research based on field experiments that conducted on paddy-wheat cultivation land with three fertilization rates. The experimental results showed that the growth space of paddy-wheat root is mainly restricted to a cylinder with a diameter of 180 mm and height of 200 mm at the seedlings stage. There were slow fluctuating trends in growth by the mean values of five new indicators within a single volume of soil. The fluctuation of five new indicators was indicated in each sampling time, which decreased gradually with time. Furthermore, treatment of N70 and N130 could similarly impact root spatial heterogeneity. Therefore, we concluded that the five new indicators could quantify the spatial dynamics of the root system of paddy-wheat at the seedling stage of cultivation. It is of great significance to the comprehensive quantification of crop roots in targeted breeding programs and the methods innovation of field crop root research.

## Introduction

Root systems are high-value targets for ensuring plant productivity because of their potential to extract water and nutrients through a complex interaction with soil biogeochemical properties [1], which helps maintain these functions under a wide range of stress scenarios and hold their

**Data Availability Statement:** All relevant data are within the paper and its Supporting Information files.

**Funding:** This work was supported by the National Natural Science Foundation of China (Grant number 31901455) and the Natural Science Foundation of Jiangsu (Grant number BK20180534) that were obtained by Xinxin Chen. The funder played a main role in study design, decision to publish and preparation of the manuscript. There was no additional external funding received for this study.

**Competing interests:** The authors have declared that no competing interests exist.

ability to anchor the plant [2–8]. Root systems are also considered essential components of ecosystems since their net primary productivity (NPP) accounts for 40% to 88% of the total NPP [9]. The spatial distribution of the root system in soil (RSA) affects the functions of the root system. These highly affected factors are anchoring, storage, transport, acquiring spatially variable resources (mobile and immobile), and competition for space, water, and nutrients [10].

RSA usually represents the morphological and structural distribution of the root [6] and describes the in-situ space-filling properties of a root system within the soil. In this case, RSA implies the global geometric configuration, which could be divided into geometric properties such as the axis root growth angle between the root growth direction and the horizontal plane and topological properties, like root branching pattern [11, 12]. Temporal and spatial patterns of water and nutrient acquisition [6, 13] and anchorage are determined by the extension rates of the individual root axes. The formation of field crops RSAs is determined by plant genetics and characteristics of the soil environment, such as water and nutrient availability, soil porosity, and rhizosphere size [14, 15]. Hence, quantifying or knowing the parameters of crop RSAs in different soil conditions is crucial for selecting optimal varieties under adverse conditions [16, 17]. However, it is limited by growing conditions and research methods.

Lab-based methods are usually performed to determine root architectural parameters such as root length, root numbers, root growth angles, root density at various depths, and root length in different diameter classes [18–21] with manual or automatized image processing [22]. One can directly derive the parameters of the root system [16, 23–26] from the lab-based methods. Nevertheless, laboratory or greenhouse conditions can differ considerably from natural conditions such as precipitation, radiation, wind patterns, soil structure, and soil depth [17, 27–29]. And the root system will have a plastic response to its growing environment [30, 31]. Also, the significant difference in the growth environment can lead to ignoring the extraction of some vital information when quantifying root architecture characteristics. Thus, this study focuses on field crops.

For field crops grown in opaque soil environments, the RSAs are complicated to observe and quantify [32]. Many efforts have been made to digitize the field crop root system. The trench profile method in which the vertical and horizontal root distribution of crops could be digitization by root length, root number, root dry mass [33], limited the quantification of root system to the profile wall. The auger method could evaluate the root distribution only in narrow horizontal planes [34, 35], which primarily used to quantify vertical root distribution with root length, root number, root dry mass [36]. The minirhizotron method has been used to estimate root growth dynamics including root turnover, root diameter, root hair [37, 38]. The monolith method is also traditionally used for root samplings, in which the size of a monolith is more significant than that of a cylinder in soil core sampling, for instance, a cylinder of diameter 50 cm was used for maize root sampling for root length, root surface area, root structure [39, 40]. The shovelmics method is a simple root sampling method, which uses shovel to dig roots [41]. It is therefore not suitable for the study of root distribution in soil, but is mainly used for root branch estimation and measurement of root growth angle of crowns [41, 42].

The abovementioned field crop indicators reflect the local, overall, plane, spatial, single, and group growth characteristics of roots, which are still insufficient to comprehensively quantify root traits. The soil heterogeneity manifests itself in depth and horizontal directions, including circumferential direction, which affects root growth. The quantification of root distribution, however, is mainly related to soil depth but rarely concerns circumferential orientation. Thus, this study analyzes the root system's ability to search circumferentially for soil growth space with a series of new indicators from field RSAs visualization technology for paddy-wheat under different N application rates during sowing. Our goal is to provide new content for the improvement of the quantitative index of the root architecture system.

## Materials and methods

### Site description

The field experiments were conducted near Babaiqiao (118˚59'E, 31˚98'N) at Nanjing, Jiangsu, Eastern China. The climate here belongs to the subtropical monsoon climate, with a mean annual temperature of 15.1˚C and sufficient sunlight. The experimental area locates in the middle and lower reaches of the Yangtze River, with an annual average rainfall of about 1000 mm and high groundwater levels. In the field, the rice-wheat rotation period is long, and a relatively hard plow bottom layer has formed at a depth of 10–12 cm soil layer due to alternating floods and droughts and mechanized operation. Sand, silt, and clay content in the soil layer from 0 to 10 cm are 3.9%, 81.5%, and 14.6%, respectively. The soil organic matter, total N, total P, total K, available P, available K, and pH are 13.91 g·kg⁻¹, 1.12 g·kg⁻¹, 0.66 g·kg⁻¹, 12.19 g·kg⁻¹, 18.18 mg·kg⁻¹, 237.25 mg·kg⁻¹ and 5.6 respectively.

### Experimental design

The wheat root was selected as the research object in rice-wheat rotation system, which was named paddy-wheat root. Wheat seed (Ningmai13) was grown in a paddy field after harvesting rice, and surface straw was manually removed prior to sowing [25]. The wheat seed was sown in untilled soil with a plant spacing of 15 mm and row spacing of 200 mm in small plots ($5 \times 3$ m²) in 3 replications and the machine drainage ditches were opened between plots on November 6, 2019. Then, sowing ditches with a width of 40–50 mm and a depth of 30–50 mm were manually opened along the row direction. Fertilizer was applied along the sowing ditches and spaced 50 mm from the sown ditches (Fig 1). Before sowing, the application of diammonium phosphate was 375 kg·hm², potassium chloride was 375 kg·hm², and the applications of urea were 70 kg·hm², 100 kg·hm² and 130 kg·hm², which represented the three N application

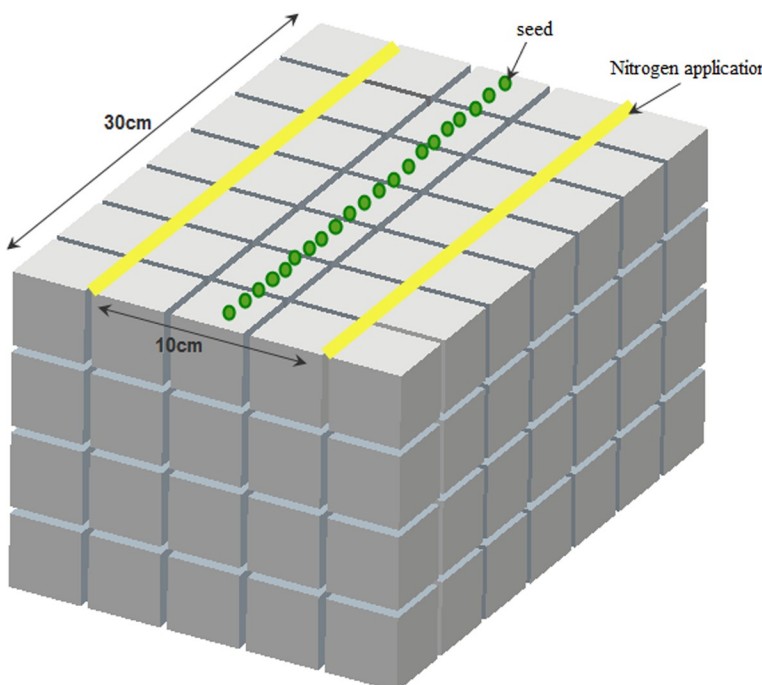
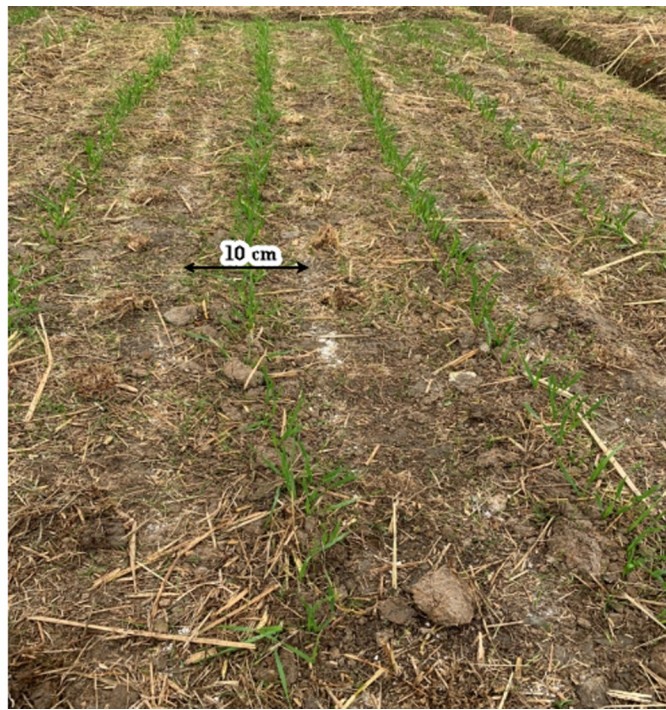

**Fig 1. Fertilization and field experiment design.**

rates and were named N70, N100, and N130 respectively. No irrigation was applied during the growing seasons due to the abundant rainfall. Herbicides and pesticides were sprayed according to standard growing practices to avoid yield loss.

## Structure of wheat root system

The root system of wheat consists of both seminal and adventitious roots. Germinating caryopsis has seminal roots (also called primary roots) that develop at the scutellar and epiblast nodes of the embryonic hypocotyl. Adventitious roots emerged from coleoptile nodes at the base of the apical culm and tillers [43]. The seminal and adventitious roots of wheat are referred to as axis roots mentioned in this text, which require digitization and visualization (Fig 2). Axial root functions of wheat mainly include the uptake and transport of soil resources, such as water and nutrients, the framework for later root development and connection, storage of photosynthetic products, and physical support of the shoot system. In addition, the root growth systems are hierarchical, and axis roots are the carrier of the lower order class of root

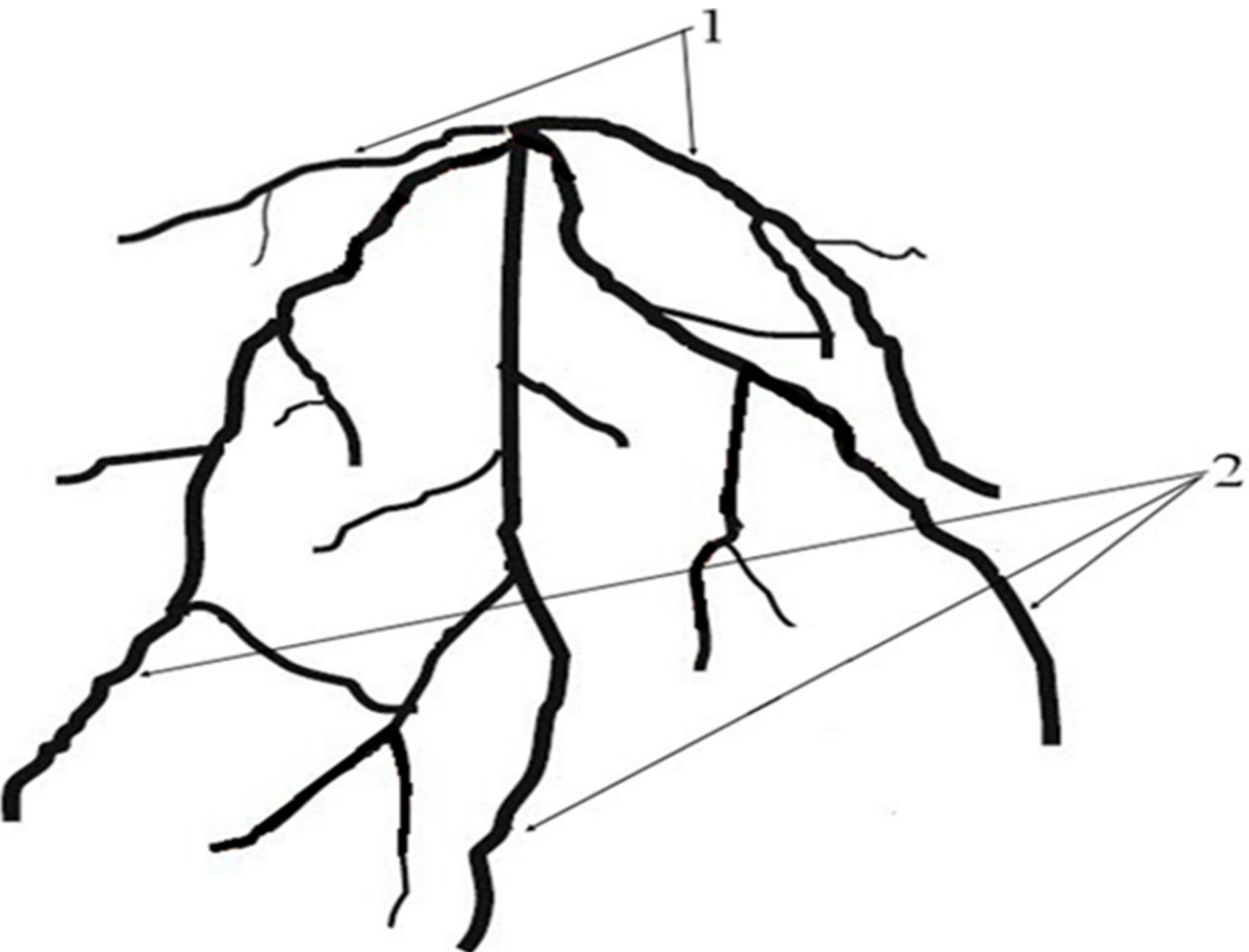

**Fig 2. Structure of wheat root system.** (1) Primary root. (2) Adventitious root.

members [44], which determines the framework of root architecture. This set the size and architecture requirements of the axis roots, making the detection and quantification of the spatial configuration of the axis roots an important research object.

## Visualization of wheat root system

Root zone samples of plants with the average appearance in the field were collected every 14 days from sowing until the 70th day. Four plant roots were excavated in a plot of each N application rate to reduce spatial heterogeneity in the soil, so twelve samples were collected in each sampling period for three N application rates. A large soil core of 180 mm in diameter and 250 mm in height with a wheat root system placed concentric at the base of the plant stem was excavated with a shovel and then brought to the laboratory for digitization [25, 41]. The data of the wheat root system (refer as wheat axis roots) were collected in a layered excavating procedure as described by Chen et al. [45] with an adapted digitizer and then transferred the data to Pro-E for 3D visual modeling.

## Circumferential distribution parameterization of wheat axis roots

The virtual visualization of cylindrical soil space surrounding root system with seed point as the origin was carried out by Pro-E after the visualization of root system (Fig 3A), then the soil column was evenly divided into eight soil spaces (S1, S2, S3, S4, S5, S6, S7 and S8) (Fig 3B) along the circumferential direction which contains eight vertical projection planes (V1, V2, V3, V4, V5, V6, V7 and V8) (Fig 3C) and eight horizontal projection planes (F1, F2, F3, F4, F5, F6, F7 and F8) (Fig 3F), from which a set of indices including proportion of single root system on eight vertical projection planes (PRVP), convex hull area of single root system on eight vertical projection planes (CHARVP), width depth ratio of roots on eight vertical projection plane (WDRRVP), length of single root system in eight soil space (RLSS) and proportion of single root system on eight horizontal projection planes (PRHP) were provided to characterize the spatial distribution characteristics of paddy-wheat root system and the dynamics of root system architectures (RSAs) with time at seedling stage under different nitrogen application rates. These five indicators are defined as follows, and all could be calculated and obtained through PRO-E analysis module

**Length of the single root system in eight soil spaces (RLSS).** The root length of eight sections was calculated using the calculation module of PRO-E (Fig 3B). The dynamics of root length in eight soil spaces were consistent with the dynamics of root length density, as the volume of each part was equal, representing trends in root water, fertilizer, and nutrient uptake.

**Proportion of the single root system on eight vertical projection planes (PRVP).** The roots were projected into eight vertical projection planes. The projection plane was divided into 50×18 small squares with a side length of 5 mm, which formed a rectangle with a length of 250 mm and a width of 90 mm [25]. The ratio of the number of grids containing wheat roots to the total number of grid cells was defined as the capacity of wheat roots for search for immovable nutrients along the vertical direction (Fig 3C).

$$P = \frac{n}{50 \times 18} \times 100\%$$

where n was the number of grids containing root sections, p was PRVP by root.

**Convex hull area of the single root system on eight vertical projection plane (CHARVP).** Points 8 mm horizontally or vertically from the root seed and the convex point of root sections on each vertical projection plane were taken as the point to form the envelope

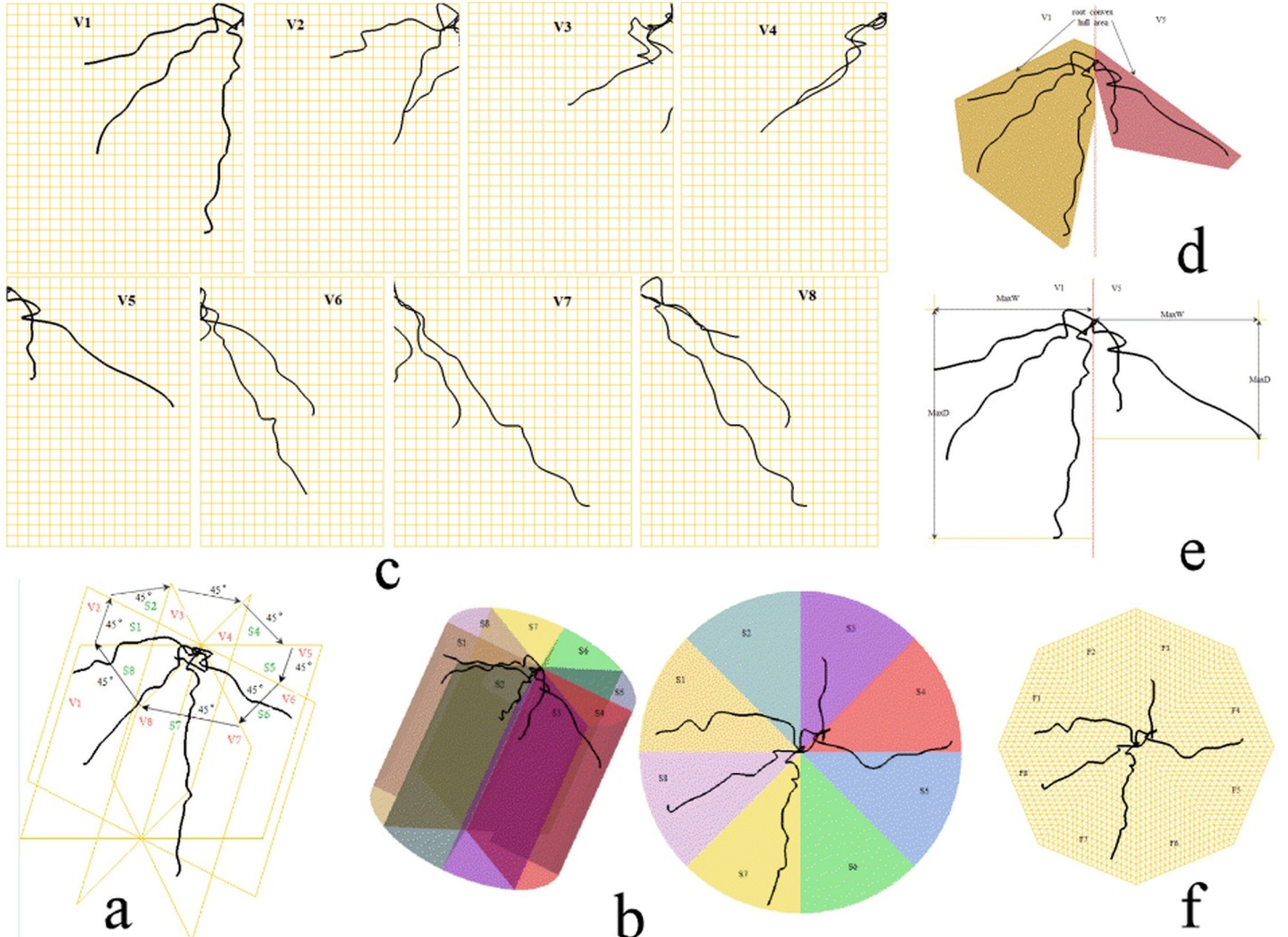

**Fig 3. Quantification of root traits in Pro-E.** (a) RSA in Pro-E. (b) the soil space that contain RSA divided along circumferential orientations. (c) the measurement of PRVP. (d) the measurement of CHARVP. (e) the measurement of WDRRVP. (f) the measurement of PRHP.

surface after projection of the root system, which was calculated using PRO-E calculation module (Fig 3D).

**Width depth ratio of the single root system in eight soil spaces (WDRRVP).** The width was prescribed as the maximum horizontal expansion between the seed and root section on the vertical projection plane, and the determination of root maximum depth on the vertical projection plane was a reference to the method proposed by Clark et al. [16], which also can be calculated by PRO-E (Fig 3E). Then the width depth ratio was the ratio between the maximum width and the maximum depth.

**Proportion of single root system on eight horizontal projection planes (PRHP).** The root system that is projected onto the horizontal projection plane consists of eight parts. Each part was divided into 20×20 small isosceles triangles with an apex angle of 45° and waist length of 5 mm, which form an isosceles triangle with an apex angle of 45° and waist length of 100 mm. And the ratio between the number of grids containing roots and the total number of grids was defined as the ability of wheat roots to search for immovable nutrients

along the horizontal direction (Fig 3F).

$$P = \frac{n}{20 \times 20} \times 100\%$$

where n was the number of grids containing root sections, p was PRHP by root.

## Results and discussions

### 3D root models of paddy-wheat

RSAs status and characteristics at different time nodes during the wheat growth period can be intuitively displayed by crop root visualization technology based on soil core, axis root digitization, and Pro-E. This method was used to reconstruct the 3D root models with time under three N application rates, as shown in Fig 4. The axis RSA could represent the root system framework [46], and the 3D models of axis roots could represent the paddy-wheat roots appropriately, as stated previously. The lines shown in Fig 4 visualize the dynamics of roots in the horizontal and vertical planes. The wheat grown in paddy soil had significant dispersal characteristics in RSAs from the horizontal planes and previous studies [47]. The RSAs of field paddy-wheat showed a highly diverse pattern even under the same fertilization treatment. In paddy-wheat, the changes in RSAs have mainly reflected the increase in axis roots and the rise in distribution within circumferential soil space. The reason could be the root emergence and disappearance with soil depth during different sampling periods by spatial heterogeneity of soil and alternation of roots, i.e., new roots emerged and old ones died. Moreover, the growth space of paddy-wheat root was a cylinder shape with a diameter of 180 mm and height of 200 mm until the 70th day after sowing (Fig 4), which is in line with previous studies [48, 49] for anaerobic and impervious plowing layer formation in rice-wheat rotation. Besides, the sampling depth for root dry weight also reported that 80–90% of the total root dry weight was distributed in the top 0–20 cm of soil layer [50].

### Traits of the paddy-wheat roots

The average value is the commonly used statistical data to show the characteristics of each observation value relative to the data set, which is used to reflect the general level of the phenomenon. Therefore, the total root length of wheat, the average length of the roots in a single spatial soil (ARLSS), the average convex hull area of roots on a single vertical projection plane (ACHARVP), the average proportion of roots on a single vertical projection plane (APRVP), the average width depth ratio of roots on a single vertical projection plane (AWDRRVP) and the average proportion of roots on a single horizontal projection plane (APRHP) were obtained by averaging the value of root length, RLSS, CHARVP, PRVP, WDRRVP and PRHP. As shown in Fig 5, there was a fluctuating rise for root length and a high variability for other traits. The means and standard deviations of the traits for all the seedlings within 70 days were $101.963 \pm 72.490$ mm for ARLSS, $3918.594 \pm 1516.143$ mm$^2$ for ACHARVP, $0.064 \pm 0.018$ for APRVP, $0.899 \pm 0.448$ for AWDRRVP, $0.071 \pm 0.048$ for APRHP when the amount of fertilizer was 70 kg.hm$^2$ (N70), $103.639 \pm 83.358$ mm for ARLSS, $4349.469 \pm 1628.500$ mm$^2$ for ACHARVP, $0.067 \pm 0.020$ for APRVP, $0.742 \pm 0.371$ for AWDRRVP, $0.068 \pm 0.054$ for APRHP when the amount of fertilizer was 100 kg.hm$^2$ (N100), and $102.716 \pm 72.726$ mm for ARLSS, $4063.723 \pm 1586.109$ mm$^2$ for ACHARVP, $0.065 \pm 0.018$ for APRVP, $0.748 \pm 0.287$ for AWDRRVP, $0.071 \pm 0.049$ for APRHP when the amount of fertilizer was 130 kg.hm$^2$ (N130). Although the distribution characteristics and growth trend of paddy-wheat roots in soil spaces under three nitrogen application rates were different, the changing trend was relatively stable.

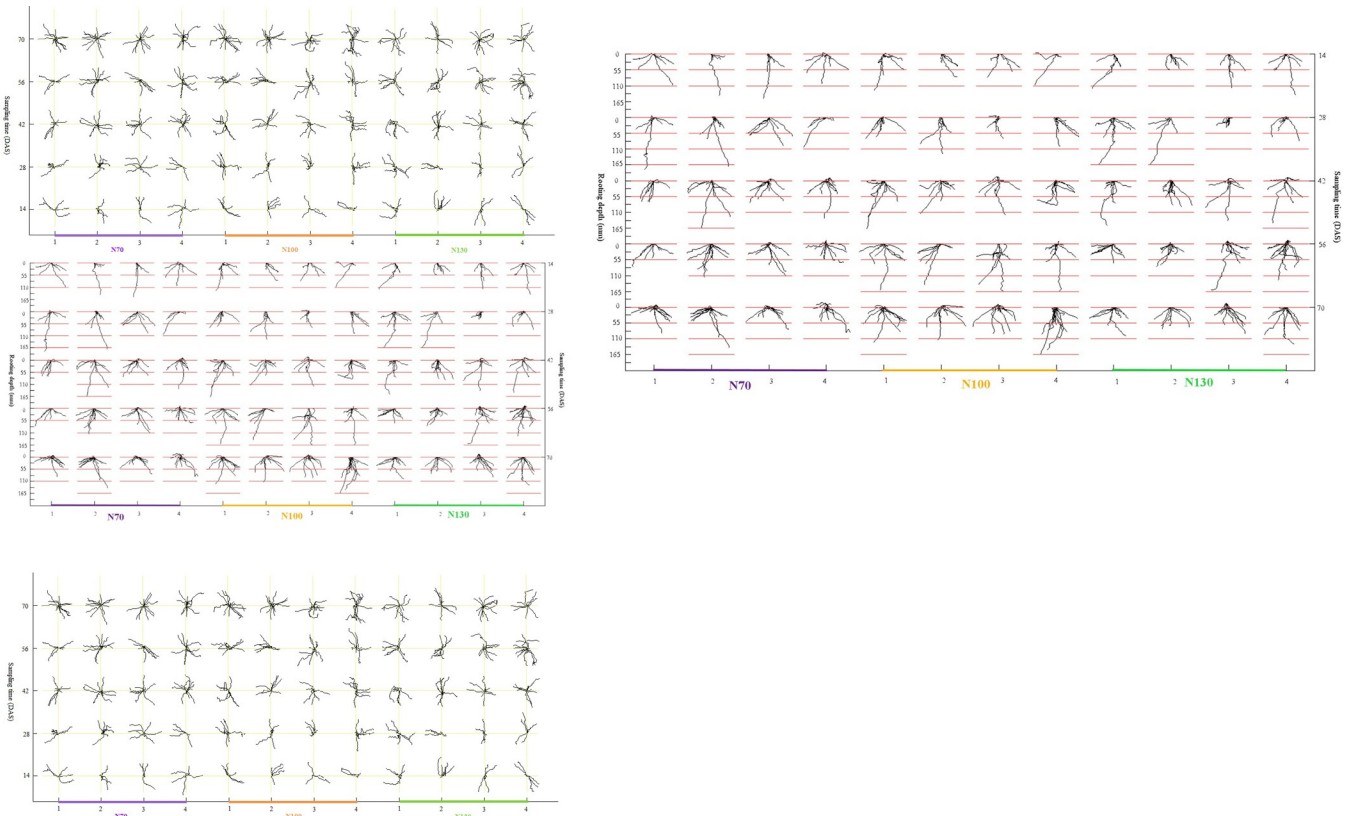

**Fig 4. 3D models of the seeding roots for different fertilization with time.**

This phenomenon may be consistent with the notion that nitrogen application rates had more effect on lateral root growth than axis root [51–53].

## Root spatial distribute quantification of paddy-wheat along circumferential soil directions

Root ability to explore the growth space was different even growing in a homogeneous environment under controlled conditions [54], making root spatial distribute quantification of field paddy-wheat along circumferential soil directions necessary. Therefore, five new indicators (PRVP, CHARVP, WDRRSP, PRHP, and RLSS) derived from the combination of root visualization and common indices were proposed, calculated, and displayed in Fig 6 (see 6A1, 6B1, 6C1, 6D1 and 6E1). Color differences marked all samples during the same sampling period. Differences among samples with changes in N application rates were analyzed using the above indexes with standard deviation (Fig 6A2, 6B2, 6C2, 6D2 and 6E2). The descriptive statistics of the above indices were listed in Table 1. A one-factorial ANOVA of samples was carried out, and the difference was marked with the mean value by an asterisk ($p < 0.05$) or double asterisk ($p < 0.01$), respectively.

Crop root is mainly used for the uptake of nutrients from the soil, and efficient root uptake of fixed nutrients such as phosphate is confined to within mm scales [55, 56]. So, the PRVP is proposed to reflect the potential ability of the paddy-wheat root to search for immovable resources in the soil space, which can be parameterized through meshing analysis on the projected RSA in the 2D viewport, in which the RSA projection has meshed in a 2.5 mm

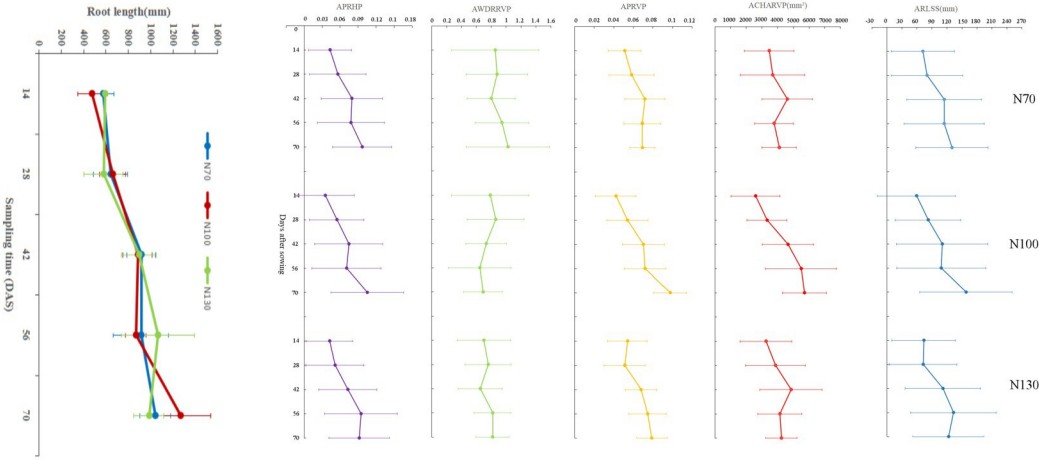

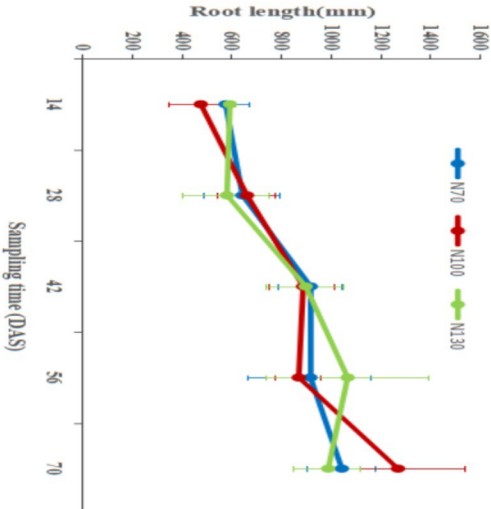

**Fig 5. Root traits dynamics.**

resolution. The PRVP calculated with Pro-E illustrated root growth distribution along with circumferential directions by dividing the soil space into eight parts instead of vertical obtained by monoliths or trenches. The differences in PRVP were reflected among samples, the changes in N application rate and growth time (Fig 6A1). The dispersion between samples decreases in a fluctuating pattern with the increase of time (Fig 6A2). The change in PRVP value indicated that the potential of field paddy-wheat roots to search for immobile soil nutrients in different circumferential soil spaces would change with time and space (Table 1).

The convex hull area and width depth ratio characterize geometric features and root sizes [57, 58]. So, the geometrical characteristics and sizes of field paddy-wheat roots in soil space (Fig 6B1, 6B2, 6C1 and 6C2) quantified with the CHARVP and the WDRRVP, which extended to imply the search-ability of wheat roots for soil, water and fertilizer along the circumferential soil space. Previous studies have usually reflected these two-dimensional indicators related to crop yield [59], phosphorus uptake [60], and field anchorage strength [61]. Illustrating how environmental factors in different directions contribute to the major mechanisms affecting

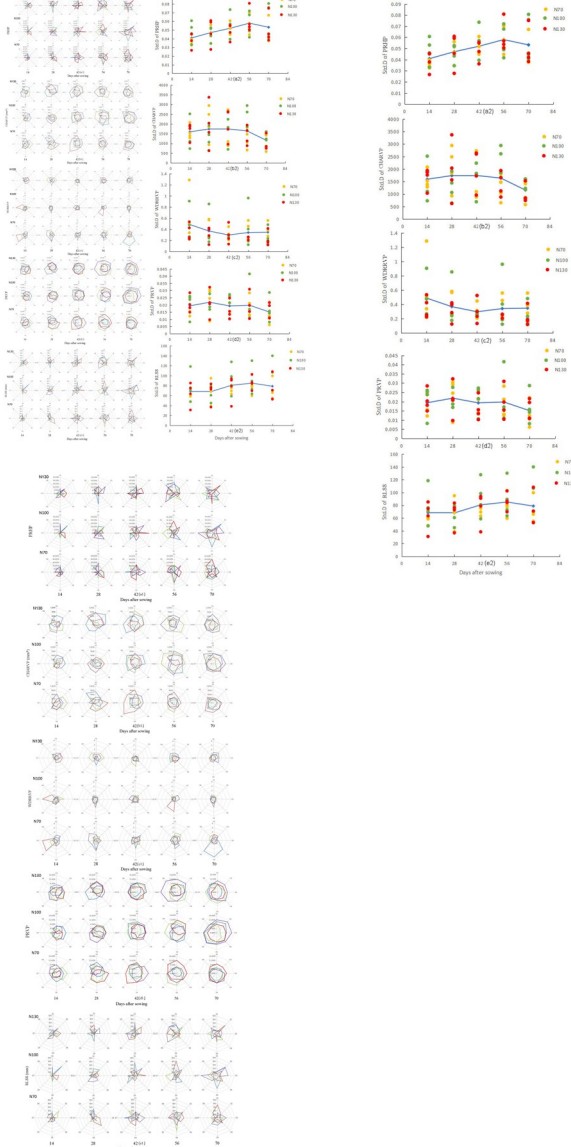

**Fig 6. Growth distribution dynamics of paddy-wheat root system in soil space under different nitrogen application rates.**

crop RSAs processes could be beneficial. The fluctuation of the observed CHARVP and WDRRVP means that periodic infiltration or the dynamics of water and fertilizer distribution among soil spaces alters the sideway elongation potential of the root system, whose mechanism still awaits further investigation.

The PRHP parameterized the foraging capacity of the roots in each horizontal orientation. This is supported by previous research [25] suggesting that the ratio of roots to horizontal projection planes be used to quantify the root potential for searching immobile nutrients in soil space along horizontal directions. As for PRHP, there were abrupt root changes in paddy-wheat (Fig 6D1). Axial root distribution and nutrient absorption capacity were more different along circumferential directions than the above three indices. The standard deviation of PRHP increased initially and then decreased throughout the seedling stage (Fig 6D2 and Table 1).

**Table 1. Descriptive statistics of paddy-wheat root traits in soil space along circumferential direction with different nitrogen applications during the sampling time.**

| | | N70 | | | | | N100 | | | | | N130 | | | | |
|---|---|---|---|---|---|---|---|---|---|---|---|---|---|---|---|---|
| | | 14 | 28 | 42 | 56 | 70 | 14 | 28 | 42 | 56 | 70 | 14 | 28 | 42 | 56 | 70 |
| PRVP | Min | 0.027 | 0.026 | 0.04 | 0.044 | 0.05 | 0.013 | 0.023 | 0.033 | 0.043 | 0.069 | 0.023 | 0.021 | 0.041 | 0.048 | 0.053 |
| | Max | 0.076 | 0.093 | 0.105 | 0.098 | 0.09 | 0.07 | 0.083 | 0.098 | 0.102 | 0.122 | 0.083 | 0.081 | 0.09 | 0.103 | 0.1 |
| | Mean | 0.051* | 0.058 | 0.072 | 0.07** | 0.069* | 0.042 | 0.054 | 0.07 | 0.072 | 0.098** | 0.054 | 0.051** | 0.068 | 0.075 | 0.079* |
| | Med | 0.049 | 0.058 | 0.071 | 0.065 | 0.07 | 0.041 | 0.053 | 0.073 | 0.071 | 0.099 | 0.054 | 0.049 | 0.069 | 0.076 | 0.082 |
| | Std.D | 0.018 | 0.025 | 0.022 | 0.02 | 0.014 | 0.022 | 0.023 | 0.023 | 0.023 | 0.018 | 0.022 | 0.023 | 0.017 | 0.03 | 0.017 |
| | CV* | 0.357 | 0.454 | 0.305 | 0.297 | 0.207 | 0.555 | 0.436 | 0.33 | 0.345 | 0.194 | 0.426 | 0.45 | 0.261 | 0.28 | 0.218 |
| CHARVP(mm$^2$) | Min | 1160.9 | 1184.5 | 1916.1 | 2140.6 | 2499.5 | 552.8 | 1417.8 | 1795 | 2419.4 | 3651.3 | 893.9 | 1273.1 | 1978.1 | 2264.6 | 2822.3 |
| | Max | 6121.2 | 7205.6 | 6992.3 | 5850.5 | 5827.2 | 4943.1 | 5368.9 | 6774.44 | 8854.4 | 7980.4 | 5513.2 | 6712.7 | 7842.6 | 6469.9 | 5782.5 |
| | Mean | 3453 | 3673.3 | 4607* | 3764* | 4095** | 2582.8 | 3319.1* | 4651.5 | 5494.4* | 5699** | 3249.5* | 3856** | 4849.1 | 4132** | 4232** |
| | Med | 3259 | 3495.4 | 4749.5 | 3375.9 | 4163 | 2410.3 | 3321.1 | 5016.5 | 5569 | 5459.3 | 3170.3 | 3681.8 | 4467.3 | 4211.4 | 4216.8 |
| | Std.D | 1679.7 | 2193.5 | 1725.4 | 1334.6 | 1171. | 1665 | 1357.2 | 1755.5 | 2427.1 | 1499.9 | 1770.9 | 2032.5 | 2119.2 | 1499.8 | 1055.8 |
| | CV* | 0.49 | 0.6 | 0.36 | 0.37 | 0.28 | 0.65 | 0.43 | 0.37 | 0.48 | 0.27 | 0.57 | 0.54 | 0.43 | 0.37 | 0.26 |
| WDRRSP | Min | 0.3 | 0.39 | 0.44 | 0.44 | 0.48 | 0.22 | 0.39 | 0.42 | 0.22 | 0.4 | 0.27 | 0.28 | 0.31 | 0.49 | 0.46 |
| | Max | 1.879 | 1.66 | 1.38 | 1.58 | 1.95 | 1.88 | 1.48 | 1.29 | 1.42 | 1.23 | 1.34 | 1.29 | 1.21 | 1.19 | 1.12 |
| | Mean | 0.852 | 0.88** | 0.8 | 0.94 | 1.02 | 0.78 | 0.86 | 0.73 | 0.65 | 0.69 | 0.7 | 0.76 | 0.65 | 0.82** | 0.82** |
| | Med | 0.565 | 0.78 | 0.68 | 0.84 | 0.73 | 0.66 | 0.75 | 0.64 | 0.48 | 0.61 | 0.61 | 0.72 | 0.54 | 0.79 | 0.81 |
| | Std.D | 0.628 | 0.44 | 0.34 | 0.38 | 0.6 | 0.55 | 0.41 | 0.29 | 0.45 | 0.27 | 0.38 | 0.33 | 0.32 | 0.26 | 0.24 |
| | CV* | 0.694 | 0.55 | 0.44 | 0.41 | 0.5 | 0.68 | 0.46 | 0.41 | 0.61 | 0.4 | 0.54 | 0.46 | 0.48 | 0.34 | 0.28 |
| RLSS(mm) | Min | 2.65 | 5.47 | 21.81 | 6.06 | 23.86 | 0.86 | 1.26 | 9.14 | 19.28 | 40.63 | 7.2 | 4 | 25.77 | 31.5 | 40.12 |
| | Max | 189.80 | 222.95 | 260.72 | 257.97 | 257.8 | 234.4 | 199.47 | 294.2 | 294.54 | 326.62 | 177.11 | 194.05 | 249.39 | 281.12 | 260.36 |
| | Mean | 71.48 | 79.92 | 114.32 | 114.16 | 129.93 | 58.99 | 82.26 | 110.5 | 108.25 | 158.19 | 73.65 | 72.17 | 111.78 | 132.87 | 123.11 |
| | Med | 48.66 | 54.73 | 105.82 | 103.47 | 136.98 | 29.64 | 64.72 | 82.77 | 80.09 | 136.26 | 53.89 | 50.27 | 101.25 | 120.21 | 104.71 |
| | Std.D | 67.60 | 76.33 | 80.07 | 85.88 | 77.59 | 83.70 | 69.93 | 97.83 | 95.2 | 98.9 | 68.09 | 71.97 | 80.77 | 91.66 | 76.26 |
| | CV* | 0.97 | 1.01 | 0.72 | 0.78 | 0.59 | 1.44 | 0.85 | 0.88 | 0.89 | 0.61 | 0.92 | 1.01 | 0.72 | 0.73 | 0.65 |
| PRHP | Min | 0.002 | 0.004 | 0.006 | 0.006 | 0.021 | 0 | 0.001 | 0.004 | 0.01 | 0.008 | 0.004 | 0.001 | 0.014 | 0.016 | 0.033 |
| | Max | 0.106 | 0.143 | 0.177 | 0.167 | 0.169 | 0.146 | 0.136 | 0.182 | 0.191 | 0.065 | 0.105 | 0.144 | 0.152 | 0.189 | 0.178 |
| | Mean | 0.043 | 0.056 | 0.079 | 0.078 | 0.097 | 0.035 | 0.054 | 0.074 | 0.071 | 0.032 | 0.042 | 0.051 | 0.073 | 0.095 | 0.092 |
| | Med | 0.032 | 0.041 | 0.074 | 0.072 | 0.102 | 0.014 | 0.05 | 0.066 | 0.056 | 0.03 | 0.026 | 0.038 | 0.064 | 0.089 | 0.078 |
| | Std.D | 0.038 | 0.051 | 0.054 | 0.06 | 0.052 | 0.052 | 0.049 | 0.061 | 0.062 | 0.02 | 0.041 | 0.052 | 0.052 | 0.065 | 0.054 |
| | CV* | 0.923 | 0.975 | 0.713 | 0.81 | 0.535 | 1.5 | 0.923 | 0.819 | 0.88 | 0.6 | 0.98 | 1.03 | 0.715 | 0.707 | 0.617 |

Med represents median, Std.D represents Std.deviation, CV

* represents coefficient of variation

** represents p < 0.01,* represents p < 0.05.

One distinctive feature of PRHP showed the spatial distribution of individual crop roots along with eight horizontal directions instead of the planar distribution obtained by core sampling or trenching.

Root length is a critical index to describe the morphological characteristics of crop roots [62] that fixes soil volume, is associated with plant efficiency in nutrient acquisition [63], and has also been linked to drought tolerance under field conditions at the seedling stage [64, 65]. The RLSS describes the growth and distribution capacity of the root system of a single crop along circumferential directions. The RLSS explains the potential of root exploitation in parallel with different orientations and is thus an indicator between the priorities of root foraging and the soil environment (e.g., nutrient distribution, water availability, or spatial heterogeneity

of soil physical properties). Similarly to the PRHP dynamic, RLSS was found to vary significantly along different directions (Fig 6E1 and 6E2). As the growth time increased, the spatial distribution of roots in each soil increased, and the spatial heterogeneity declined.

These above indicators are quite different from the commonly used indicators (root angle, root length, root mass, etc.), which describe the changes in root characteristics along with the soil depth. They are also different from indices describing local growth characteristics of roots, such as root hair and root branch [66], and indices describing global root growth characteristics, such as the fractal dimension of roots [67]. They are derived from the combination of previously used quantitative indicators and a new analysis perspective based on visualization technology of the field paddy-wheat root system. These indicators can enable us to quantify roots more comprehensively than before.

## Effect of nitrogen application rate on root circumferential spatial distribution

Regression analysis and one-factorial ANOVA were conducted on the dynamics of each indicator's mean and coefficient of variation (CV) for the paddy-wheat root system under three nitrogen application rates. The regression equations and correlation coefficients were obtained, as shown in Table 2, where X1 representative N70, X2 representative N100, and X3 representative N130. For the five indicators proposed, there was a linear relationship between different nitrogen application rates; however, the coefficient of determination was quite different. Compared with the mean value dynamics of five indicators, changes in RLSS and PRHP were similar; The coefficients of determination ($R^2$) between N70 and N100 were higher than that between N100 and N130 and between N70 and N130, were 0.89, 0.61, and 0.86 for the RLSS, and 0.96, 0.71, and 0.83 for the PRHP respectively. The correlation of the CHARVP between N70 and N100 ($R^2 \approx 0.3$) and between N100 and N130 ($R^2 \approx 0.48$) was relatively lower than that between N70 and N130 ($R^2 \approx 0.9$). Otherwise, the correlation of the PRVP between N100 and N130 was higher ($R^2 \approx 0.82$) than that between N70 and N100 ($R^2 \approx 0.67$) and between N70 and N130 ($R^2 \approx 0.71$). The correlation between N70 and N130 ($R^2 \approx 0.84$) was higher than that between N70 and N100 ($R^2 \approx 0.24$) and between N100 and N130 ($R^2 \approx 0.18$) for the WDRRVP. One-factorial ANOVA pointed out significant differences ($p < 0.05$) between N100 and N70, and between N100 and N130 for the WDRRVP. For the CV dynamics of five indicators, the correlation between N70 and N130 was all higher than that between N70 and N100, and between N100 and N130.

The treatment of N70 and N130 had a similar impact on spatial heterogeneity of roots, except for WDRRVP dynamics (Tables 1 and 2). This result may be supported by previous research in which the appropriate amount of nitrogen fertilizer will play a positive role in the roots' development of cotton [68]. The results from one-factorial ANOVA for samples of different sampling times (Table 1) showed no significant difference for the RLSS and PRHP within 70 days and for the PRVP, CHARVP, and WDRRVP at some time nodes. Thus, even though fewer data were available, these new indicators that represent variability in root growth variability along the circumferential direction, which reflect space plasticity of the root system, are needed to improve the quantitative evaluation system of crop roots because root growth dynamics comprehensively embodies adaptation ability to multiple factors [69].

## Conclusions

Five new indicators for phenotypic quantification of paddy-wheat roots' spatial distribution along circumferential direction were provided, which were used to quantify the dynamics of the root system architecture (RSA) along its eight-part circumferential orientations with

**Table 2. Correlation of various dynamics of circumferential root characteristics under different fertilization rates.**

| fertilization rates | | indicators | regression equations | $R^2$ |
|---|---|---|---|---|
| | Mean | RLSS | X2 = 1.4×X1-39.14 | 0.892 |
| | | PRHP | X2 = 1.205×X1-0.017 | 0.961 |
| | | CHARVP | X2 = 1.668×X1-2185.39 | 0.303 |
| N70 and N100 | | PRVP | X2 = 1.934×X1-0.056 | 0.671 |
| | | WDRRVP | X2 = -0.473×X1+1.167 | 0.243* |
| | CV | RLSS | X2 = 1.18×X1-0.025 | 0.448 |
| | | PRHP | X2 = 1.338×X1-0.105 | 0.5 |
| | | CHARVP | X2 = 0.611×X1+0.171 | 0.298 |
| | | PRVP | X2 = 0.56×X1+0.175 | 0.573 |
| | | WDRRVP | X2 = -0.548×X1+0.21 | 0.233 |
| | Mean | RLSS | X3 = 0.594×X2+41.145 | 0.609 |
| | | PRHP | X3 = 0.76×X2+0.019 | 0.709 |
| | | CHARVP | X3 = 0.296×X2+2775.32 | 0.479 |
| | | PRVP | X3 = 0.53×X2+0.03 | 0.82 |
| N100 and N130 | | WDRRVP | X3 = -0.37×X2+1.023 | 0.178 |
| | CV | RLSS | X3 = 0.275×X2+0.514 | 0.294 |
| | | PRHP | X3 = 0.398×X2+0.407 | 0.509 |
| | | CHARVP | X3 = 0.673×X2+0.129 | 0.57 |
| | | PRVP | X3 = 0.69×X2+0.067 | 0.782 |
| | | WDRRVP | X3 = -0.29×X2+0.254 | 0.117 |
| | Mean | RLSS | X3 = 1.044×X1-3.756 | 0.855 |
| | | PRHP | X3 = 1.009×X1-0.001 | 0.829 |
| | | CHARVP | X3 = 1.228×X1-747.47 | 0.897 |
| | | PRVP | X3 = 1.163×X1-0.009 | 0.708 |
| N70 and N130 | | WDRRVP | X3 = 0.769×X1+0.057 | 0.838* |
| | CV | RLSS | X3 = 0.871×X1+0.092 | 0.946 |
| | | PRHP | X3 = 0.97×X1+0.042 | 0.84 |
| | | CHARVP | X3 = 0.897×X1+0.054 | 0.809 |
| | | PRVP | X3 = 1.063×X1-0.016 | 0.842 |
| | | WDRRVP | X3 = 0.559×X1+0.121 | 0.336 |

X1 represent N70, X2 represent N100, X3 represent N130

* represent $0.01 < p < 0.05$.

visualization technology consisting of in situ field root samplings, RSA digitization, and reconstruction. These proposed indicators reflected the ability of the root to explore circumferential soil space, either directly or indirectly, which are essential for quantitative RSA description. Also, this research has shortcomings, such as fewer samples for each N application treatment each time, only one year of the experiment, and only one wheat variety. However, these proposed indicators in this research could promote comprehensive quantification of field crop RSAs or provide more 3D information for roots by linking them to the laboratory phenotyping platform.

## Supporting information

**S1 Data. Data for Figs 4–6.**
(XLSX)

**S2 Data. Data for Tables 1 and 2.**
(XLSX)

## Acknowledgments

We are grateful to Ruiyin He and Qishuo Ding, the professors of College of Engineering, Nanjing Agricultural University, who provided necessary help during experiment implementation in the field. And also we are grateful to Khurram Yousaf, the assistant professor of National University of Sciences and Technology Islamabad, who helped to revise the manuscript.

## Author Contributions

**Conceptualization:** Qingfei Duan.

**Funding acquisition:** Xinxin Chen.

**Investigation:** Xinxin Chen, Yongli Tang, Qingfei Duan.

**Methodology:** Xinxin Chen.

**Resources:** Xinxin Chen.

**Software:** Qingfei Duan.

**Supervision:** Jianping Hu.

**Validation:** Yongli Tang, Qingfei Duan, Jianping Hu.

**Visualization:** Xinxin Chen, Qingfei Duan.

**Writing – original draft:** Xinxin Chen, Yongli Tang.

**Writing – review & editing:** Jianping Hu.

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
