## [Decision Letter · Decision Letter 0]

17 Aug 2022

PONE-D-22-15992Phenotyping for circumferential expansion of field paddy-wheat root system architecturePLOS ONE

Dear Dr. Hu,

Thank you for submitting your manuscript to PLOS ONE. After careful consideration, we feel that it has merit but does not fully meet PLOS ONE’s publication criteria as it currently stands. Therefore, we invite you to submit a revised version of the manuscript that addresses the points raised during the review process.

 The reviewers raised some concerns about the manuscript that must be addressed. Authors are advised to carefully revise the manuscript and address all the concerns of the reviewers and submit a revised version.

We look forward to receiving your revised manuscript.

Kind regards,

Academic Editor

PLOS ONE

Journal Requirements:

Additional Editor Comments:

The reviewers raised some concerns about the manuscript that must be addressed. Authors are advised to carefully revise the manuscript and address all the concerns of the reviewers and submit a revised version.

Reviewers' comments:

Reviewer's Responses to Questions

**Comments to the Author**

1. Is the manuscript technically sound, and do the data support the conclusions?

Reviewer #1: Yes

Reviewer #2: Partly

Reviewer #3: Yes

2. Has the statistical analysis been performed appropriately and rigorously? 

Reviewer #1: Yes

Reviewer #2: Yes

Reviewer #3: Yes

3. Have the authors made all data underlying the findings in their manuscript fully available?

Reviewer #1: Yes

Reviewer #2: Yes

Reviewer #3: Yes

4. Is the manuscript presented in an intelligible fashion and written in standard English?

Reviewer #1: Yes

Reviewer #2: Yes

Reviewer #3: No

5. Review Comments to the Author

Reviewer #1: The manuscript has been wisely put together with very clear vision and results. The use of figures is very intelligent. Overall it reflects a strong scientific research article.

The manuscript has a clear description of results with extensively defined figures.

Reviewer #2: overall the manuscript is well written but major revision requires. it will be accepted after revision. NO novelty statement was nor provided and some statistical error in your manuscript. result and material method was written well.

Reviewer #3: The present manuscript entitled as “Phenotyping for circumferential expansion of field paddy-wheat root system architecture” describes the findings of a field experiment on wheat under three fertilization rates and provided some new parameters to quantify the dynamics of the root system architecture along its orientations divided into eight parts with visualization technology that consists of root in situ samplings. Authors collected plant samples during growing season 2019-20. The research idea and objectives are unique and research methodology is well organized according to the objectives. However, there are some flaws and drawbacks in the present version of article.

The manuscript is poorly written, there are some sentences which are not meaningful or don’t make any clear sense for the readers. So, I recommend that the current version of the submitted manuscript should be very carefully revised and upgraded. After critical and major revision in the writings, I recommend to accept this manuscript. I also, have some other major concerns which are appended below:

Line 4: you used the word “hidden half” in abstract, does it indicates roots? If so then it’s better to use roots to make the sentence easy to understand.

Line 11: It is not clear whether you conducted research on paddy and wheat or on wheat alone that was grown in paddy soil, make it clear. You didn’t mention Nitrogen fertilizer rate used in you treatments?

Lines 11-12: You mentioned in manuscript that “we provided five new to quantify the dynamics of the root system architecture”; what did you provide new?

Line 20: you didn’t mention nitrogen rate and you stated about N70 and N130 nitrogen treatment in your results, provide treatment details clearly.

Line 49-51: clearly state about “angle”.

Line 63: what do you mean by “Depth50 and Width50”?

Line 89-90: Provide country name as well. Replace light with sunlight. Provide groundwater level if possible.

Line 93-94: Make this sentence grammatically correct.

Line 134-135: You didn’t mention three nitrogen rate in methodology, provide clear details of your treatments?

Line 179: Check and correct spellings

Line 201: Check and correct the formula.

Line 234: Use superscript for mm and do same for all manuscript.

Line 255-256: Provide space between words and parenthesis, do same for all manuscript.

Line 351: What new indicators did you observe, write those.

6. PLOS authors have the option to publish the peer review history of their article (what does this mean?). If published, this will include your full peer review and any attached files.

Reviewer #1: No

Reviewer #2: **Yes: **

Reviewer #3: No

---

## [Author Response · Author response to Decision Letter 0]

4 Oct 2022

Dear editor,

Thank you very much for providing me the opportunity to revise the manuscript.

We have revised the above-mentioned manuscript carefully according to the comments from the Journal and reviewers. The responses to the Journal’ requirements and reviewer’s comments and changes made are listed as follows:

Journal’s Requirement 1:

Responses and Changes Made:

According to the PLOS ONE's style requirements, we have changed the article structure consist of abstract, introduction, materials and methods, results, discussion into abstract, introduction, materials and methods, results and discussion, conclusion. And the title_authors_affiliations were also modified in title page as required.

Journal’s Requirement 2:

In your Data Availability statement, you have not specified where the minimal data set underlying the results described in your manuscript can be found. PLOS defines a study's minimal data set as the underlying data used to reach the conclusions drawn in the manuscript and any additional data required to replicate the reported study findings in their entirety. All PLOS journals require that the minimal data set be made fully available. For more information about our data policy, please see http://journals.plos.org/plosone/s/data-availability.

Responses and Changes Made:

According to the PLOS ONE's data requirements, we rewritten it in the part of the materials and methods marked with blue font (Line 135-138), also we rearranged all the relevant data and put them in the Supporting Information files (S1 Data).

Reviewer #1

Comment of reviewer 1: 

The manuscript has been wisely put together with very clear vision and results. The use of figures is very intelligent. Overall it reflects a strong scientific research article.

The manuscript has a clear description of results with extensively defined figures.

Responses and Changes Made:

Thanks very much for the comments of the reviewer. We will continue to work hard to present our future studies in a scientific, understandable and wise way. We hope readers in this field or other fields all can get some useful information from it.

Reviewer #2

Comment 1 of Reviewer 2:

overall the manuscript is well written but major revision requires. it will be accepted after revision. NO novelty statement was nor provided and some statistical error in your manuscript. result and material method was written well.

The manuscript titled “Phenotyping for circumferential expansion of field paddy-wheat root systemarchitecture” emphasize on wheat root system with reference to circumferential orientations. Using three fertilization rates and experiment was designed to quantify dynamic root system architecture (RSA). With field root in situ samplings and RSA digitization, authors findings are presented in this manuscript. 

Introduction and material method section is written well. 

In result section table-2, last column is not understandable, either it is p-value or t-value. If it is p-value, that is non-significant and if it is p-value that is also non-significant. Without clarification of this aspect, this table is not able to be published. 

Responses and Changes Made:

According to the reviewer’s comments, we revised Table 2 marked in blue font and listed the average value and coefficient of variation of each indicator as the comparison object, then we constructed the relationship function between different nitrogen application rates. The P value was not listed, but where there were significant differences, the asterisk (p<0.05) or double asterisk (p<0.01) was used to mark the corresponding R² value, and we put the data in the Supporting Information files (S2 Data). 

Comment 2 of Reviewer 2

Variation has not been well explained in reference to p-value, confidence level etc; so under these condition, such results are unacceptable. Author should justify the results with p-value using analysis of variance models. 

Responses and Changes Made:

According to the reviewer’s comments, we analyzed the novel indicators with p-value using One-factorial ANOVA and justified the results in the part of Results and discussions marked with blue font (Line363-370 ). 

Comment 3 of Reviewer 2

In line # 243, author mentioned one-factorial anova, but I could not see that in table with p-value. 

Responses and Changes Made:

According to the reviewer’s comments, we illustrated p<0.05 and p<0.01 with an asterisk and double asterisk, respectively ( Table 1 and Table 2).

Comment 4 of Reviewer 2

The overall conclusion of the study is missing and describe the novelty of your research .Under such conditions, this study is not publishable unless the above mentioned suggestions are addressed properly. 

Responses and Changes Made:

According to the reviewer’s comments, we supplemented the conclusion in which we described the novelty of our research in the part of the Conclusions marked with blue font (Line 376-393). 

Reviewer #3

Comment 1 of Reviewer 3:

The present manuscript entitled as “Phenotyping for circumferential expansion of field paddy-wheat root system architecture” describes the findings of a field experiment on wheat under three fertilization rates and provided some new parameters to quantify the dynamics of the root system architecture along its orientations divided into eight parts with visualization technology that consists of root in situ samplings. Authors collected plant samples during growing season 2019-20. The research idea and objectives are unique and research methodology is well organized according to the objectives. However, there are some flaws and drawbacks in the present version of article.

The manuscript is poorly written, there are some sentences which are not meaningful or don’t make any clear sense for the readers. So, I recommend that the current version of the submitted manuscript should be very carefully revised and upgraded. After critical and major revision in the writings, I recommend to accept this manuscript. I also, have some other major concerns which are appended below:

Responses and Changes Made:

According to the reviewer’s comments, the manuscript has been revised upgraded carefully, especially for writings. We also discussed it with a foreign assistant professor to reduce its writing problems and improve the readability. We did not to mark the above changes with colored fonts, because the whole article has been modified. For other concerns, we have made corresponding modifications and marked with blue font.

Comment 2 of Reviewer 3:

Line 4: you used the word “hidden half” in abstract, does it indicates roots? If so then it’s better to use roots to make the sentence easy to understand.

Responses and Changes Made:

According to the reviewer’s comments, we have revised "hidden half" with roots in the abstract for the word “hidden half” in abstract indicates roots, and the modified part was marked with blue font (Line 4). 

Comment 3 of Reviewer 3:

Line 11: It is not clear whether you conducted research on paddy and wheat or on wheat alone that was grown in paddy soil, make it clear. You didn’t mention Nitrogen fertilizer rate used in you treatments?

Responses and Changes Made:

According to the reviewer’s comments, to clarify this problem, we illustrated it in the parts of the Materials and methods (Line 100-103) that we conducted the research on wheat alone grown in paddy soil. For the Nitrogen fertilizer rate used in our treatments, we also revised it in the parts of the Materials and methods (Line 108-111). And the modified part was marked with blue font.

Comment 4 of Reviewer 3:

Lines 11-12: You mentioned in manuscript that “we provided five new to quantify the dynamics of the root system architecture”; what did you provide new?

Responses and Changes Made:

According to the reviewer’s comments, we concluded it in the part of the Results and Discussions (Line 327-335) and Conclusions (Line376-388) marked with blue font. In short, the new indicators come from a new analysis angle combined with previous root research methods and quantitative indicators.

Comment 5 of Reviewer 3:

Line 20: you didn’t mention nitrogen rate and you stated about N70 and N130 nitrogen treatment in your results, provide treatment details clearly.

Responses and Changes Made:

According to the reviewer’s comments, we supplied the treatments of nitrogen fertilizer rates in detail in the parts of the Materials and methods marked with blue font (Line 108-111), shown as “Before sowing, the application of diammonium phosphate was 375 kg∙hm2, potassium chloride was 375 kg∙hm2, and the applications of urea were 70 kg∙hm2, 100 kg∙hm2 and 130 kg∙hm2, which represented the three N application rates and were named N70, N100, and N130 respectively.”

Comment 6 of Reviewer 3:

Line 49-51: clearly state about “angle”.

Responses and Changes Made:

According to the reviewer’s comments, we changed the angles into the root growth angles (Line 49-51) stated clearly in Line 38-41, which marked with blue font. The angle used in the manuscript implies the root growth angle between the root growth direction and the horizontal plane. 

Comment 7 of Reviewer 3:

Line 63: what do you mean by “Depth50 and Width50”?

Responses and Changes Made:

According to the reviewer’s comments, we revised this problem and deleted it because it indicates a lack of use. The Depth50 and Width50 represent the vertical and horizontal centroids of root distribution, respectively [Teramoto S and Uga Y, 2020].

Comment 8 of Reviewer 3:

Line 89-90: Provide country name as well. Replace light with sunlight. Provide groundwater level if possible.

Responses and Changes Made:

According to the reviewer’s comments, we provided the country name and replaced light with sunlight marked with blue font (Line 89-90). However, we did not find the groundwater level of this area in the authoritative data, so we did not supplement the data on the groundwater level.

Comment 9 of Reviewer 3:

Line 93-94: Make this sentence grammatically correct.

Responses and Changes Made:

According to the reviewer’s comments, we changed this sentence into “In the field, the rice-wheat rotation period is long, and a relatively hard plow bottom layer has formed at a depth of 10-12 cm soil layer due to alternating floods and droughts and mechanized operation.” Marked with blue font (Line 92-95).

Comment 10 of Reviewer 3:

Line 134-135: You didn’t mention three nitrogen rate in methodology, provide clear details of your treatments?

Responses and Changes Made:

According to the reviewer’s comments, we provided the three nitrogen fertilizer rate in detail in the parts of the Materials and methods marked with blue font (Line 108-111).

Comment 11 of Reviewer 3:

Line 179: Check and correct spellings

Responses and Changes Made:

According to the reviewer’s comments, we checked and corrected the “Convsex ” with “Convex” marked with blue font (Line 181).

Comment 12 of Reviewer 3:

Line 201: Check and correct the formula.

Responses and Changes Made:

According to the reviewer’s comments, we checked and corrected the “* ” with “x” in the formula (Line 203).

Comment 13 of Reviewer 3:

Line 234: Use superscript for mm and do same for all manuscript.

Responses and Changes Made:

According to the reviewer’s comments, we used superscript for mm and checked and did the same for all the manuscript.

Comment 14 of Reviewer 3:

Line 255-256: Provide space between words and parenthesis, do same for all manuscript.

Responses and Changes Made:

According to the reviewer’s comments, we provided space between words and parenthesis, and checked and did the same for all the manuscript.

Comment 15 of Reviewer 3:

Line 351: What new indicators did you observe, write those.

Responses and Changes Made:

According to the reviewer’s comments, we introduced them in the part of the Materials and methods (Line 166-204) and concluded them in the part of the Abstract (Line 11-22) and the Conclusions (Line 376-393). Changes made above were marked with blue font. We provided five new indicators (PRVP, CHARVP, WDRRSP, PRHP and RLSS) to quantify the dynamics of the root system architecture (RSA) along its eight-part circumferential orientations with visualization technology. And the visualization technology consists of in-situ field root samplings, RSA digitization, and reconstruction according to previous research. In short, the new indicators come from a new analysis angle combined with previous root research methods and quantitative indicators.

---

## [Decision Letter · Decision Letter 1]

2 Nov 2022

PONE-D-22-15992R1Phenotyping for circumferential expansion of field paddy-wheat root system architecturePLOS ONE

Dear Dr. Jianping Hu, Thank you for submitting your revised manuscript to PLOS ONE. Reviewers are optimistic about your manuscript, however some minor revisions are still required before the manuscript can be accepted for publication. Therefore, we invite you to submit a revised version of the manuscript having addressed the points raised during the review process along with a rebuttal letter. Looking forward to receive the revised manuscript soon.

We look forward to receiving your revised manuscript.

Kind regards,

Sudeshna Bhattacharjya, Ph.D

Academic Editor

PLOS ONE

Journal Requirements:

Additional Editor Comments (if provided):

Dear Jianping Hu

Reviewers' comments:

Reviewer's Responses to Questions

**Comments to the Author**

1. If the authors have adequately addressed your comments raised in a previous round of review and you feel that this manuscript is now acceptable for publication, you may indicate that here to bypass the “Comments to the Author” section, enter your conflict of interest statement in the “Confidential to Editor” section, and submit your "Accept" recommendation.

Reviewer #2: All comments have been addressed

Reviewer #4: All comments have been addressed

2. Is the manuscript technically sound, and do the data support the conclusions?

Reviewer #2: Yes

Reviewer #4: Yes

3. Has the statistical analysis been performed appropriately and rigorously? 

Reviewer #2: Yes

Reviewer #4: Yes

4. Have the authors made all data underlying the findings in their manuscript fully available?

Reviewer #2: Yes

Reviewer #4: Yes

5. Is the manuscript presented in an intelligible fashion and written in standard English?

Reviewer #2: Yes

Reviewer #4: Yes

6. Review Comments to the Author

Reviewer #2: appreciation for authors that they incorporate all the objection carefully. This is strong enough to meet the potential to be published. i strongly recommended the publication of this paper

Reviewer #4: • I have gone through the whole manuscript very thoroughly and found that the author has written it very extensively and descriptively.

• The title should be reframed as it looks like a simple sentence. It should be catchy, attractive and reflective of the research hypothesis.

• The abstract was written very scantily. It should be required. Therefore, please rewrite the abstract with proper hypotheses, the origin of study with clear-cut objectives, a brief methodology, and finally, results with proper justification and clarity.

• The amount of rainfall received during the experimental period was important in determining crop growth and development, and no irrigation was provided during the growing seasons. You have mentioned that plant roots are essential for water and nutrient absorption, anchoring, mechanical support, metabolite storage, and interaction with the surrounding soil environment. As a result, I believe that water budgeting is necessary to justify the hypothesis because water is an important factor that influences root growth and development in field conditions.

• Figure quality may be improved for more clarity.

• The conclusion is very condensed; there is scope to improve it briefly as per the salient achievement of the study. Hence, it can be elaborated with final recommendations.

7. PLOS authors have the option to publish the peer review history of their article (what does this mean?). If published, this will include your full peer review and any attached files.

Reviewer #2: No

Reviewer #4: **Yes: **Dr Bharat Prakash Meena

---

## [Author Response · Author response to Decision Letter 1]

14 Nov 2022

Dear editor,

Thank you very much for providing me the opportunity to revise the manuscript again.

We have revised the above-mentioned manuscript carefully according to the comments from the Journal and reviewers. The responses to the Journal’ requirements and reviewer’s comments and changes made are listed as follows:

Journal Requirements:

Responses and Changes Made:

According to the Journal’s requirements, we reviewed our reference list and found that the reference No. 68 has gone on the journal’s website (Chen J, Wang P, Ma ZM, Lyu XD, Liu TT, Siddique KHM. Optimum water and nitrogen supply regulates root distribution and produces high grain yields in spring wheat (Tritium aestivum L.) under permanent raised bed tillage in arid northwest China. Soil Till Res. 2018; 181:117-126.). So we removed and replaced it with a relevant current reference, also modified the corresponding context in the manuscript marked with the blue font (L361-363, Line 580-583). 

Reviewer #2

Comment of reviewer 2: 

Appreciation for authors that they incorporate all the objection carefully. This is strong enough to meet the potential to be published. i strongly recommended the publication of this paper

Responses and Changes Made:

Thanks very much for the comments and valuable suggestions of the reviewer for improving this manuscript. We are all glad that our manuscript revision has satisfied the reviewer.

Reviewer #4: 

Comment 1 of Reviewer 4:

I have gone through the whole manuscript very thoroughly and found that the author has written it very extensively and descriptively.

Responses and Changes Made:

We agree with the reviewer’s comments, and we hypothesize that the indicators describing the characteristics of root system architecture still need to be supplemented. Taking the root system of field paddy-wheat as the research object, we propose five new indicators characterizing the root spatial distribution characteristics along the circumferential directions based on the previous research methods. And the test results show that the newly proposed indicators can quantitatively describe the characteristics of spatial root distribution along the circumferential directions. However, as we mentioned in the conclusion, fewer samples, only one year of the experiment, and only one wheat variety are shortcomings in writing it very affirmatively and specifically.

Comment 2 of Reviewer 4:

The title should be reframed as it looks like a simple sentence. It should be catchy, attractive and reflective of the research hypothesis.

Responses and Changes Made:

According to the reviewer’s comments, we changed the title “Phenotyping for circumferential expansion of field paddy-wheat root system architecture” to “Phenotypic quantification of root spatial distribution along circumferential directions for field paddy-wheat” after carefully thinking and discussing.

Comment 3 of Reviewer 4:

The abstract was written very scantily. It should be required. Therefore, please rewrite the abstract with proper hypotheses, the origin of study with clear-cut objectives, a brief methodology, and finally, results with proper justification and clarity.

Responses and Changes Made:

According to the reviewer’s comments, we rewrote the abstract, supplied the hypotheses (Lines 7-8), and refined the goals and methods (Lines 11-16), all marked with blue font. We framed the abstract as background (Lines 2-6), hypotheses and illustrations (Lines 7-11), goals and methods (Lines 11-16), results (Lines 16-22), conclusions and prospects (Lines 22-25).

Comment 4 of Reviewer 4:

The amount of rainfall received during the experimental period was important in determining crop growth and development, and no irrigation was provided during the growing seasons. You have mentioned that plant roots are essential for water and nutrient absorption, anchoring, mechanical support, metabolite storage, and interaction with the surrounding soil environment. As a result, I believe that water budgeting is necessary to justify the hypothesis because water is an important factor that influences root growth and development in field conditions.

Responses and Changes Made:

Thanks for the reviewer’s comments; Nanjing locates in the middle and lower reaches of the Yangtze River, with an annual average rainfall of about 1000 mm and high groundwater levels. The growth of field wheat here mainly depends on rainfall without irrigation. And this study mainly proposes new quantitative indicators of wheat root growth distribution growing in natural conditions. The test data used belonged to the same growth cycle, so we did not consider the impact of rainfall on root growth during the experiment. Also, we did not record rainfall data or soil moisture content during sampling time. According to the reviewer’s comments, we searched the rainfall data from the government's website and list them from 2010-2019 in the below chart (http://www.jiangsu.gov.cn/col/col76741/index.html), for the rainfall data of 2020 was not published. And the data marked with red font covered the sampling time. We think it may not be essential to supply this part in the manuscript. 

Year Jan Feb Mar Apr May Jun Jul Aug Sep Oct Nov Dec Total

2010 18.8 115.6 117.8 197.9 56.1 62.1 343.3 142.1 181.1 32.1 7.6 24 1298.4

2011 10.8 17.2 43.2 11.6 40.6 312.9 278 284.3 12.6 28.7 21.3 15.8 1077

2012 21 73.3 79.3 56.2 62.4 17.8 176.4 198.3 68.7 54.6 42.8 66.4 917.2

2013 17.8 69.9 42.9 22.8 110.1 172.6 229.6 115.5 67 22.4 17.2 10.6 898.4

2014 20.6 121.2 68.4 97.6 26.3 111.8 263.5 158.8 89.2 32 97.9 3.8 1091.1

2015 29.9 58.1 104.8 121.6 96.1 661.5 258 187.4 63.6 61.6 110.7 12.3 1765.6

2016 62.8 31.1 40 155.4 119.6 186.2 477.3 78.7 187.4 308 95.4 65.8 1807.7

2017 59.3 38.8 55.1 117.9 83.9 309.1 99.6 217.1 176.4 81 7.4 9.5 1255.1

2018 106 44.9 120.7 61 142.8 49.1 182.3 273.6 66.2 37.5 65.6 117.4 1267.1

2019 51.5 90.4 37.6 51.5 41.1 109.2 63.9 123 51.4 1.4 41.7 59.1 721.8

Comment 5 of Reviewer 4:

Figure quality may be improved for more clarity.

Responses and Changes Made:

The figures uploaded are compressed according to the requirements of the journal. According to the reviewer’s comments, we uploaded clearer pictures in supporting information.

Comment 6 of Reviewer 4:

The conclusion is very condensed; there is scope to improve it briefly as per the salient achievement of the study. Hence, it can be elaborated with final recommendations.

Responses and Changes Made:

According to the reviewer’s comments, we rearranged the conclusion marked with blue font (Lines 377-386).

---

## [Decision Letter · Decision Letter 2]

6 Dec 2022

Phenotypic quantification of root spatial distribution along circumferential direction for field paddy-wheat

PONE-D-22-15992R2

Dear Dr. Hu,

We’re pleased to inform you that your manuscript has been judged scientifically suitable for publication and will be formally accepted for publication once it meets all outstanding technical requirements.

Kind regards,

Sudeshna Bhattacharjya, Ph.D

Academic Editor

PLOS ONE

Additional Editor Comments (optional):

Reviewers' comments:

Reviewer's Responses to Questions

**Comments to the Author**

1. If the authors have adequately addressed your comments raised in a previous round of review and you feel that this manuscript is now acceptable for publication, you may indicate that here to bypass the “Comments to the Author” section, enter your conflict of interest statement in the “Confidential to Editor” section, and submit your "Accept" recommendation.

Reviewer #4: All comments have been addressed

2. Is the manuscript technically sound, and do the data support the conclusions?

Reviewer #4: Yes

3. Has the statistical analysis been performed appropriately and rigorously? 

Reviewer #4: Yes

4. Have the authors made all data underlying the findings in their manuscript fully available?

Reviewer #4: Yes

5. Is the manuscript presented in an intelligible fashion and written in standard English?

Reviewer #4: Yes

6. Review Comments to the Author

Reviewer #4: References should be in uniform style, so there is a need to check the style and format of references in the text as well as in the references list.

Conclusion is written well although there is scope to improve it in briefly as per salient achievement of the study. It should be in bullet information.

7. PLOS authors have the option to publish the peer review history of their article (what does this mean?). If published, this will include your full peer review and any attached files.

Reviewer #4: No

---

## [Editor Report · Acceptance letter]

10 Jan 2023

PONE-D-22-15992R2 

Phenotypic quantification of root spatial distribution along circumferential direction for field paddy-wheat 

Dear Dr. Hu:

I'm pleased to inform you that your manuscript has been deemed suitable for publication in PLOS ONE. Congratulations! Your manuscript is now with our production department. 

Kind regards, 

on behalf of

Dr. Sudeshna Bhattacharjya 

Academic Editor

PLOS ONE